# Effects of Complex Rehabilitation Program on Reducing Pain and Disability in Patients with Lumbar Disc Protrusion—Is Early Intervention the Best Recommendation?

**DOI:** 10.3390/jpm12050741

**Published:** 2022-05-02

**Authors:** Emilian Tarcău, Dorina Ianc, Elena Sirbu, Doriana Ciobanu, Ioan Cosmin Boca, Florin Marcu

**Affiliations:** 1Department of Physical Education, Sport and Physiotherapy, University of Oradea, 410087 Oradea, Romania; emilian.tarcau@yahoo.com (E.T.); dorina.ianc@yahoo.com (D.I.); doriana.ciobanu@yahoo.com (D.C.); icboca@yahoo.com (I.C.B.); 2Human Performance Research Center, University of Oradea, 410087 Oradea, Romania; 3Department of Physical Therapy and Special Motricity, Faculty of Physical Education and Sport, West University of Timisoara, 300223 Timisoara, Romania; 4Department of Psychoneuroscience and Rehabilitation, University of Oradea, 410087 Oradea, Romania; mfmihai27@yahoo.com

**Keywords:** low back pain, physical therapy, disability, hydrotherapy

## Abstract

(1) Background: Due to its frequency and possible complications, low back pain (LBP) has a high social impact, it is a common problem of the active population and the second reason for visiting a physician. In patients with lumbar disc protrusion (LDP), one of the most common causes of LBP, the nucleus pulposus bulges against the disc and then protrudes into the spinal cord, but the annulus fibrosus remains intact; (2) Objectives: The primary objective of this study was to determine the efficacy of a rehabilitation treatment (RT) comprising electrotherapy (ET), hydrotherapy (HT) and individualized physical therapy (PT) versus ET alone in patients with LDP. The second objective was to investigate whether there is a correlation between early RT and the symptomatology of patients with LDP; (3) Methods: The research was conducted between July 2021 and January 2022 at the Ceres Hotel Treatment Centre from Baile 1 Mai, Romania, and all the study subjects signed an informed consent form. For this study, the block randomization method was used to randomize subjects into groups that resulted in equal sample size, in order to maintain a reasonably good balance among groups. Therefore, the two groups had the same number of subjects (30 subjects) and the randomization was made taking into account the patient’s motivation or the subject’s willingness to receive not only electrotherapy treatment, but also the physical exercises and hydrotherapy. The eligibility criteria were: low back pain for more than three months, an MRI confirmed diagnosis of LDP (without dural compression), and ability to perform a PT program. The control group received only a classical ET program. In addition, the patients in the experimental group received a complex individualized PT program associated with HT and ET. To achieve these objectives, the study subjects were monitored for spinal mobility (lateral lumbar flexion—LLF, index fingers-ground—IFG, lumbar Schober tests for flexion—LS, Inverted Schober test for extension—ILS), trunk flexor and extensor muscle strength (LF strength, LE strength), level of pain (Short Form McGill Pain Questionnaire—SF-MPQ, Visual Analogue Scale—VAS), and the degree of limitation in activities of daily living (Oswestry Disability Index—ODI); (4) Results: Comparing the evolution of the subjects, using the One-Way ANOVA between groups, we observed a significant improvement in all variables [SF-MPQ (95% CI, 7.996/11.671), VAS (95% CI, 1.965/3.169), mobility FTF (95% CI, −7.687/−3.940), LS (95% CI, 2.272/2.963), LE strength (95% CI, −5.676/−3.324), LF strength (95% CI, −5.970/−3.630), disability (95% CI, 8.026/10.441) after six months of treatment for the experimental group subjects. A clear correlation was found, using the Bravis–Pearson test, between the earliest possible initiation of RT and improvement of patients’ symptoms; (5) Conclusion: The current study proves the importance of combining ET with HT and PT. The earlier the RT is implemented, the lower the pain perception and level of disability associated with the lumbar disease.

## 1. Introduction

Low back pain (LBP) is a common problem in the active population, and is considered to be the second reason for visiting a physician. Besides the fact that LBP causes a substantial morbidity, being the main reason for disabilities in the active population, it also generates major expenses consisting of treatment costs, lost workdays and decreased productivity [1]. In patients with lumbar disc protrusion (LDP), the nucleus pulposus bulges against the disc and further on protrudes into the spinal column, but the annulus fibrosus remains intact. Previous studies reported a significant association between disc imaging findings including disc protrusion and LBP. Disc protrusions were found in the asymptomatic adult population, with a prevalence from 10% to 30% and increases with age. Conversely, in symptomatic patients the reported prevalence was around 40% [2,3,4]. The monthly prevalence of lumbar disc disease is estimated at 43% of the population; thus, LBP is the second reason for visiting a general practitioner or consultant [5]. In Romania, the prevalence of LBP in adults is high (62%), second only to headaches (79%) in the ranking of painful disorders [6]. The high prevalence of the lumbar disc degenerative disease is explained by the presence of several risk factors, which are more and more present in our society: weight lifting, ageing, sedentary lifestyle, smoking, obesity and genetics [7]. Studies have revealed that genetic factors may also be involved in lumbar disc disease and in other inflammatory rheumatic diseases [8,9,10].

It is known that patients with LDP prefer to start their treatment with conservative methods such as analgesics, epidural steroid injections, electrotherapy (ET) and physical therapy (PT). Lately, the updated 2021 Clinical Practice Guidelines (CPGs) for LBP focused on which intervention could be recommended for providing pain and/or disability reduction and functional improvement. Moreover, the recommendation of the 2021 LBP CPGs update focused on the type of intervention delivered by physiotherapists [11]. Fritz et al. conducted a study on 220 patients with acute LBP and compared treatment with spinal manipulation combined with trunk-strengthening exercises and education (experimental group) to usual care (control group). The experimental group registered statistically significant improvement in disability, but the improvement was modest and did not achieve the minimum clinically important difference compared to the control group [12]. Furthermore, other studies reported the benefits of manual therapy (spinal mobilization) vs. conventional physiotherapy or exercises in the management of LBP and associated disc degeneration [13,14,15,16,17,18]. Moreover, the superiority of spinal mobilizations over ET was reported [19,20]. However, the above mentioned studies do not clearly support the superiority of one type of intervention in the management of LBP. Another debate in the management of LBP is the optimum timing of non-conservative intervention after the diagnosis was established. Indeed, patients have improved outcomes in terms of pain, disability with early intervention for LBP. Moreover, early intervention is more cost-effective than delayed initiation of therapy, with delayed therapy being associated with increased cost and healthcare consumption [12]. Previous guidelines for the early management of LBP have suggested a multidisciplinary approach including education, physical activity, manual therapy, acupuncture and ET [11,21]. However, there is a lack of studies investigating the timing of interventions for LBP and LDP. Although physical therapy, traction, hydrotherapy, and electrotherapy are considered common conservative treatment methods, no unified scheme for conservative treatment has been established for LDP [22].

This study helps to improve the ways rehabilitation is provided to patients with LDP. The outcomes of this study will have a positive influence on the wellbeing of patients with LDP by helping them improve mobility and strength, and decrease the pain and disability. The rehabilitation treatment proposed may be included by physiotherapists in the routines they use with patients with LDP, and some of the physiotherapy exercises can also be included by the patients in their daily routine at home. Furthermore, the rehabilitation treatment proposed may be used by physiotherapists in other research.

The main objective of this study was to determine the effectiveness of the rehabilitation treatment (RT) versus ET-only intervention on pain and disability reduction, and the increase in muscle strength and mobility in patients with LDP. The second objective was to investigate if there is a correlation between early rehabilitation intervention and pain and disability.

## 2. Methods

### 2.1. Study Design

We performed a prospective cohort study, enrolling 60 patients (25 men and 35 women) aged between 26 to 76 years, diagnosed with LDP.

This study complied with the principles outlined in the Helsinki Declaration and was approved by the local Ethics Committee (approval no. 1947/14.07.2021). It was registered at Iranian Registry of Clinical Trails (IRCT20200422047172N2). All participants provided their written informed consent before being included in the study.

For this study, the block randomization method was used to randomize subjects into groups that resulted in equal sample size, in order to maintain a reasonably good balance among groups. Therefore, the two groups had the same number of subjects (30 subjects) and the randomization was made taking into account the patient’s motivation or the subject’s willingness to receive not only electrotherapy treatment but also the physical exercises and hydrotherapy. In this research, a single-blinded procedure was used. Given the nature of the therapeutic intervention, (e.g., exercises, electrotherapy devices, hydrotherapy), and its effects directly felt by the subjects, blinding for therapists and patients was not possible, therefore subjects were aware that they belonged to an experimental or a control group. In order to reduce bias, one team of therapists performed the therapeutic procedures, and another person who did not belong to the team performing the treatment, therefore unaware whether the patient received one type of intervention or the other, evaluated its results. 

STROBE checklist of items was respected in our study (Appendix A).

### 2.2. Setting

Patients were recruited at the Ceres Hotel Treatment Centre from Băile 1 Mai, Romania, from July 2021 to January 2022, and the study was performed over a period of six months. According to one’s personal reasons (availability, adherence, health costs etc.), the patients were assigned either to a control (*n* = 30) or experimental group (*n* = 30). The control group (Gr B) received only a classical ET program. In addition, the patients in the experimental group (Gr A) received a physical therapy (PT) program associated with hydrotherapy (HT) and ET.

The patients in the experimental group (Gr A) received a 10-day inpatient rehabilitation program, once a day. Moreover, they continued the daily physical therapy exercise program after hospital discharge.

All patients in Gr A attended the rehabilitation treatment consisting of: HT, PT and ET.

Patients were treated with 37 °C water therapy (hydrotherapy) in a therapeutic pool for 30 min a day, five-days a week for a total duration of 10 days. In Băile 1 Mai, the thermal water used for the treatment has a mineralization that is at around 1 g/L with mainly negative ions (anions)—bicarbonates, sulfates and positive ions (cations)—calcium, sodium and magnesium. Hydrotherapy is widely used for the relief of all forms of back pain [23].

A PT program was given to all patients for 45 min a day, five-days a week for a total duration of 10 days. The exercises consisted of active resisted movement of trunk flexion and extension, pelvic tilt, abdominal and trunk muscles strengthening, and trunk mobility exercises. The dosage and intensity of the exercises progressed over time. The program was individualized according to the patients’ specific impairments.

After completing the PT program, ET was applied to each patient. The ET procedures consisted of: (1) Transcutaneous electrical nerve stimulation (TENS) in conventional mode, symmetric biphasic, 90 Hz, for 15 min; (2) Interferential current (IFC) in quadripolar mode, at 100 Hz frequency, for 10 min; (3) Magnetic field therapy in continuous form (sedative effect), for 15 min.

TENS and IFC were performed using the Chattanooga Intelect Neo combined device. The magnetic field was delivered through a Physiomed MAG-Expert device with a field strength of 1–100 Gauss (adjustable in steps of one Gauss) and a frequency range from 1–100 Hz, with two completely independent channels and a treatment timer.

We used TENS and IFC in order to reduce pain. We used both types of currents to insure better reactivity and compliance from the patients. The magnetic field therapy was applied to help reduce the state of psychomotor agitation induced by pain.

After completing the 10 days program of RT in the treatment center, all patients received a written outline and description of the PT program and they were reminded to carry out the exercises regularly after discharge.

The patients in the control group were only treated with ET, for 10 days, in a hospital. The ET protocol was the same as for the experimental group.

### 2.3. Participants

The inclusion criteria were: LBP for more than three months, an MRI confirmed diagnosis of LDP (without dural compression), and ability to perform a physical therapy program. The exclusion criteria were: indication for acute surgery, previous surgery on the same lumbar spinal level, sciatica, presence of severe spinal pathology (spinal tumor, spinal fracture, spinal stenosis or radiculopathy, fibromyalgia, inflammatory and infectious spinal diseases). Chronic pain relief drug users, refusal to participate in the research, neoplasms of any sort, severe comorbidities, and mental illness are also exclusionary factors.

### 2.4. Variables

The patients presented to the rehabilitation service with the diagnosis established by the specialized doctor, based on the MRI results. In Romania, most rehabilitation centers are separate units, separate from healthcare units/hospitals, where patients come from all over the country, based on recommendations made by a specialist.

There were no cases of dropout from the therapeutic program.

### 2.5. Data Sources

The participants’ demographic characteristics such as age, gender, weight, height and body mass index (BMI) were recorded. In addition, the presence of radiculopathies and the elapsed period (in months) from diagnosis to the start of the treatment were noted.

All assessments were made before the intervention and six months later by the same physiotherapist, who was blinded to the treatment groups.

Spinal mobility was assessed using the lateral lumbar flexion (LLF) testing, index-ground (IG), lumbar Schober tests for flexion (LS), and Inverted Schober test for extension (ILS) [24,25,26,27,28].

For the muscle strength evaluation, both the strength of the flexor and extensor muscles of the trunk were tested. To assess the abdominal muscle strength, the number of trunk flexions the subject could perform in a 30 s interval was recorded. In evaluating the strength of the lumbar extensor muscles, the number of correct executions of spine extensions in a 30-s interval was noted.

Primary endpoints were the Oswestry Disability Index (ODI), the Short Form McGill Pain Questionnaire (SF-MPQ) and the pain level using Visual Analogue Scale (VAS) [29,30,31,32,33].

Secondary outcomes included muscle strength and spinal mobility evaluation using the LLF testing, IG, LS, and ILS [29].

SF-MPQ was used to determine the pain level [29,34,35]. The SF-MPQ is a multidimensional measure of perceived pain in adults with chronic pain, including pain due to lumbar disc diseases [36], and can be used to assess the interference of pain in performing the activities of daily living. The questionnaire contains a total of 15 descriptors (four affective and 11 sensory), which are rated on an intensity scale: “0” = none, “1” = mild, “2” = moderate, “3” = severe. The existing data confirm the reliability, validity, and responsiveness of SF-MPQ in patients with chronic pain [37].

Additionally, the degree of pain was evaluated using a visual analogue scale (VAS) on a 10-cm scale, with zero indicating “no pain” and 10 indicating “worst pain”.

The Oswestry Disability Questionnaire (ODQ) was used to measure the limitation in everyday life activities [38]. There is evidence supporting its validity and reproducibility [39]. The ODQ is based on 10 sections with six levels each, assessing the limitation of various activities of daily living. The values range from zero (the best health state) to 100 (the worst health state). For each section of the questionnaire, the total possible score is five. The first statement was scored zero, and consecutive statements were scored from one to five. The total score was then divided by the total possible score and expressed as a percentage to produce the Oswestry Disability Index (ODI). The ODI is interpreted as follows: 0–20%, minimal disability; 21–40%, moderate disability; 41–60%, severe disability; 61–80%, crippled; 81–100%, patients are either bed-bound or exaggerate their symptoms.

### 2.6. Bias

In order to exclude subject selection bias, we respected the natural recovery potential of LDP and excluded patients with adjuvant therapies that have a major impact influencing the ability to comply with PT. Thus, only patients with LDP of more than six months were included in this study, as the natural recovery potential decreases with increasing disease duration and will eventually approach to zero. Furthermore, those subjects who found that their ability to comply with PT had been clearly influenced by the adjuvant therapies administered were excluded during the study.

### 2.7. Study Size

In order to establish the sample size, we used as primary reference the average number of patients with low back pain who came into the clinic in a month, as being the population size. We are referring to a number of 70 patients per month. We also considered the 95% confidence level, and a margin of error of 5% and an assumed population proportion of 0.5. The z score for a 95% confidence level is 1.96. In order to obtain the sample size according to these parameters, we used an online sample size calculator. The sample size for the chosen population size is 60 subjects.

### 2.8. Quantitative Variables

For data analysis, we used the Statistical Package for Social Sciences (SPSS) Evaluation version 15.0.0, issued by IBM SPSS.Statistic, Oradea, Romania (SPSS 15.0). For the quantitative analysis of the numerical variables, we used the mean and standard deviation, and for the categorical variables we used the percentage and mean. We analyzed the normality of data distribution using the Kolmogorov–Smirnov test. For the intergroup analysis of the initial values, we used the independent samples *t*-test, as we have a normal data distribution (the Kolmogorov–Smirnov test, *p* ≥ 0.05). The Chi^2^ test for homogeneity was conducted in order to explore whether frequency counts are distributed identically across the two groups of subjects, in terms of gender and the presence of radiculopathy.

### 2.9. Statistical Methods

In order to test if there is a significant difference between the two groups for the initial and final results, we used the One-Way ANOVA between subjects, as we had a normal data distribution (the Kolmogorov–Smirnov test, *p* ≥ 0.05). For pretest–posttest analysis for the two groups we used One-Way ANOVA with repeated measures.

For the effect size measure for both One-way ANOVA between subjects and One-Way ANOVA with repeated measures, Partial Eta Squared was used.

Bravis–Pearson test was used in order to assess the correlation between the time elapsed from the diagnosis and beginning of the treatment and pain intensity; also between the time elapsed from the diagnosis and beginning of the treatment and the disability level.

In order to determine the MCID score for both SF-MPQ score and ODI score, we established a ‘cut point’ score for each outcome tool and we compared the subjects results with that ‘cut point’ score (respective 10 points improvement for SF-MPQ score and 30% improvement for ODI score). The subjects below the ‘cut point’ score were excluded and then we used a logistic regression analysis to account for the differences in sample size between models using the same process, but only including patients with an initial score above the ‘cut point’.

In total, 95% confidence intervals (CIs) were reported as appropriate.

## 3. Results

The distribution of the patients’ parameters in each group was homogeneous (differences were not statistically significant) according to the studied variables: age, gender, body mass index, time elapsed from diagnosis and start of the treatment, presence of radiculopathies (Table 1, the Kolmogorov–Smirnov test, *p* ≥ 0.05). Furthermore, there is not a significant difference between experimental and control group regarding data homogeneity according to gender [x^2^(1) = 0.61; *p* ≥ 0.05] and the presence of radiculopathy [x^2^(2) = 0.66; *p* ≥ 0.05].

There was no significant difference between the groups in terms of pain, muscle strength, mobility and disability at the initial evaluation (Table 2).

The comparison between groups of the final assessment results, show that there are significant differences regarding pain intensity in the morning (VAS-M) [F(1.59) = 11.06, *p* < 0.05]; pain intensity in the evening (VAS-E) [F(1.59) = 24.39, *p* < 0.05]; SF-MPQ pain perception [F(1.59) = 48.81, *p* < 0.05]; finger-to-floor distance [F(1.59) = 8.32, *p* < 0.05]; Schober test [F(1.59) = 26.47, *p* < 0.05]; inverted Schober test [F(1,59) = 4.73, *p* < 0.05]; ODQ score [F(1.59) = 13.15, *p* < 0.05]; ODQ (%) [F(1.59) = 9.04, *p* < 0.05]. At six months, the experimental group had improved more than the control group in regards to pain (VAS, SF-MPQ), disability (ODQ, ODQ%) and LS scores (*p* < 0.05). There was no statistically significant difference between the two groups in the trunk left lateral flexion [F(1.59) = 0.32, *p* > 0.05]; trunk right lateral flexion [F(1.59) = 0.05, *p* < 0.05]; trunk flexion strength [F(1.59) = 4.45, *p* < 0.05]; trunk extension strength [F(1.59) = 0.64, *p* < 0.05] (Table 3).

The analysis within the control group proves that there are significant differences between the pretest–posttest values for pain intensity in the morning (VAS-M) [F(1.59) = 90.26, *p* < 0.05]; pain intensity in the evening (VAS-M) [F(1.59) = 194.05, *p* < 0.05]; in the morning; SF-MPQ pain perception [F(1.59) = 48.81, *p* < 0.05]; finger-to-floor distance [F(1.59) = 356.05, *p* < 0.05]; Schober test [F(1.59) = 452.78, *p* < 0.05]; disability level [F(1.59) = 409.44, *p* < 0.05] and percentual disability level (ODQ%) [F(1.59) = 414.19, *p* < 0.05].The percentage score obtained in the assessment of disability level decreased on average by 11.93 points (95% CI [10.52, 13.34]).

The analysis within the experimental group proves that there are significant differences between the pretest–posttest values for pain intensity in the morning (VAS-M) [F(1.59) = 102.72, *p* < 0.05]; pain intensity in the evening (VAS-E) [F(1.59) = 96.91, *p* < 0.05]; SF-MPQ pain perception [F(1.59) = 124.76, *p* < 0.05]; finger-to-floor distance [F(1.59) = 49.76, *p* < 0.05]; Schober test [F(1.59) = 470.93, *p* < 0.05]; inverted Schober test [F(1.59) = 37.51, *p* < 0.05]; trunk left lateral flexion [F(1.59) = 21.19, *p* < 0.05]; trunk right lateral flexion [F(1.59) = 13.61, *p* < 0.05]; trunk forward flexion [F(1.59) = 70.38, *p* < 0.05]; trunk extension [F(1.59) = 61.27, *p* < 0.05]; disability level (ODQ) [F(1.59) = 240.78, *p* < 0.05] and the percentage of disability level (ODQ%) [F(1.59) = 307.38, *p* < 0.05]. The percentage score obtained in the assessment of disability level decreased on average by 11.93 points (95% CI [10.52, 13.34]).

Patients in the experimental group registered significant improvements in all studied variables (pain, mobility, muscle strength, disability) after six months of treatment (Table 3). This means that post-intervention, a statistically significant difference was found in the *p* values of SF-MPQ, FTF, LS, ILS, LLF, LE, ODQ (Table 3).

The data presented in bold characters highlight a statistically significant difference between groups (Inter-action) and within groups (Group A changes and Group B changes).

The analysis of the correlation between the time elapsed from diagnosis to the start of the rehabilitation treatment program and the level of pain, as well as the disability index, was performed using the Pearson correlation coefficient.

In the experimental group there is a significant positive correlation between the time elapsed from diagnosis to treatment administration, and the degree to which pain interferes with the performance of the activities of daily living r(28) = 0.81, *p* = 0.000 (Figure 1a). In the control group it is noticed that there is no significant correlation between the time elapsed from diagnosis to the start of treatment and the level of pain r(28) = 0.17, *p* = 0.930 (Figure 1b).

Regarding the correlation between the time elapsed from the diagnosis to treatment administration and the level of pain in the morning, there is no significant correlation, neither in the experimental group [r(28)= 0.047, *p* = 0.804] nor in the control group [r(28)= 0.262, *p* = 0.169]

Regarding the correlation between the time elapsed from the diagnosis to treatment administration and the level of pain in the evening, there is no significant correlation, neither in the experimental group [r(28)= 0.262, *p* = 0.169] nor in the control group [r(28)= 0.308, *p* = 0.098].

In the experimental group, we noticed a significant positive association between the elapsed period from diagnosis to the start of treatment and the disability index r(28) = 0.62, *p* = 0.000 (Figure 2a). The sooner treatment is started, the more reduced is the discomfort caused by the disease in six months of treatment. In the control group, we noticed a significant negative correlation between the time elapsed from diagnosis to treatment administration and the disability index r(28) = −0.58, *p* = 0.001 (Figure 2b). The longer the period elapsed from diagnosis to treatment, the greater the discomfort created by the disease.

## 4. Discussion

According to the Global Burden of Disease 2017 Study, the global prevalence and years lived with disability of back pain has increased between 1990 and 2017, registering the maximum prevalence in the 45–49 age group in 2017 [40].

In fact, the first degenerative changes in the disc appear after the age of 30. At the level of the annulus fibrosus there is a decrease in elasticity, while at the level of the nucleus pulposus, its hydrophilic capacity decreases, favoring the occurrence of traumatic lesions with the partial or total rupture of the annulus fibrosus, which will cause disc protrusion or disc herniation on the midline or lateral.

Patients with lumbar disc protrusion may experience back pain, reduced mobility, weakness in the legs and feet, disability during the person’s daily functioning and decreased quality of life.

According to specialized studies, the conservative treatment has proven to be successful in the treatment of patients with lumbar disc protrusion, leading to pain relief, improved mobility and symptoms severity. Over time, various methods of rehabilitation treatment in patients with LDP have been studied, including patient education, behavioral therapies, back school, exercises, PT, balneotherapy and ET (ultrasound, TENS, interferential currents, laser) [20,35,41,42].

Moreover, surgical treatment has not been shown to be more effective than conservative treatment in reducing the severity of the symptoms or improving the quality of life in patients with lumbar disc disease in the midterm or long-term follow-up as referred by Gugliotta M. et al. Although pain was relieved more quickly in patients who received surgical treatment (persistent at the three-week follow-up), the difference was no longer present after three months [43].

The results of our study show that subjects in both groups registered better outcomes after a six month rehabilitation program. However, after comparing the final parameters between the two groups, the experimental group who underwent ET, HT and PT had significantly improved in regards to pain (SF-MPQ), disability (ODQ, ODI) and LS scores.

Unlike the control group, the comparison of the initial and final results for the experimental group shows that there are significant differences between the pretest and posttest in all parameters. Thus, patients from the experimental group registered significant improvements in terms of pain and disability decrease, and mobility and muscle strength increase after six months of combined treatment.

PT refers to movement therapy and is performed by a physiotherapist. PT defines the field that studies the neuromuscular and articular mechanisms of the human body. This involves a set of programs, techniques and principles that address the human body for prophylactic, therapeutic, restorative and compensatory purposes. PT is used for functional medical rehabilitation and consists of a set of techniques and methods focused on physical exercise. The main purpose of the exercises is to regain joint mobility, regain muscle strength and the ability to exercise, improve breathing, correct coordination and muscle balance and improve the connection between the brain and muscles. If a muscle is irreversibly affected, the aim is to train other muscles to partially take over the affected person’s functions.

The benefits of ET modalities, delivered alone or in combination with other interventions, in relieving pain, improving range of motion or disability, were reported in several studies [20,42,44,45,46]. On the other hand, HT has proven beneficial effects in pain reduction and lumbar mobility in patients with LBP [39,47].

TENS therapy uses an electrical current that triggers the release of endorphins, which are the body’s natural painkillers with an analgesic effect. Moreover, TENS provides electrical stimulation through the skin, through sensory stimuli that will carry information to the brain through afferent fibers to the dorsal horn of the Aß medulla. Therefore, there is a blockage of pain impulses to the brain, as this fiber transmits the information faster than the fibers responsible for transmitting the pain through (the gate control theory) [48].

The fundamental aspects of IFC therapy involve reducing cutaneous nerve stimulation and maximizing the current that permeates the tissues with a higher carrier frequency, making it more suitable for treating deeper tissue layers [49]. Some theories have been proposed to explain the analgesic effect, such as the gate control theory, descending pain suppression pathway, physiological blockage and placebo effect [50].

The magnetic field, on the metabolic structures, determines: energetic changes at the cell surface; activation of membrane exchanges; intensification of enzymatic processes and cellular metabolism; accentuation of the repair granulation tissue; increase in vascularization in bones and scar tissue; increase in collagen synthesis capacity cartilaginous. Continuous magnetic field has predominantly anabolic effects. At the level of the central and vegetative nervous system, continuous MDF causes sedative, sympatholytic, and trophotropic effects.

Although there has been a lot of debate around the effectiveness of various interventions (alone or in combination), there are few studies in the literature that combine ET with PT and HT in treating patients with LBP [20,39,42,43,44,45,46,47].

Another finding of our study was that patients included in the experimental group showed a significant positive correlation between the time elapsed from diagnosis to treatment and the perception of pain during the day. The earlier treatment is started, the less annoying the pain is at the end of the treatment period.

Regarding the perception of pain in the morning and in the evening, evaluated using the VAS scale, we found that there is no correlation between the two moments of the day and the early start of treatment. This can be explained by the higher perception of pain in the morning due to morning stiffness [51], associated with soft tissue thixotropy, on the one hand, and exacerbated pain in the evening on the other hand, due to fatigue (physical and mental) accumulated on during the day [52].

Moreover, we noticed a significant positive association between the period elapsed from diagnosis to the start of treatment and the disability in these patients. So, we can state that the sooner the treatment starts, the more the discomfort created by the disease decreases after six months of complex treatment. In return, no positive significant correlation was observed in the control group.

For SF-MPQ, the MCID score is a single *cutpoint* that represents a change in the score (initial score minus the final score). Taking into account the samples size and the subjects score, success for the 10 points improvement models on the SF-MPQ was calculated as [SF-MPQ total score initial—SF-MPQ total score final] ≧ 10. Subjects with the baseline score already below the *cutpoint* for success were excluded and the *N* was adjusted accordingly. The dependent variable was dichotomized into successful and unsuccessful groups. A logistic regression analysis was completed to account for the differences in sample size between models using the same process, but only including patients with an initial SF-MPQ above 10 points improvement. For all regression calculations, a *p* value of ≤005 was considered significant.

In the experimental group, 43.33% from the patients registered a score over the 10-points improvement *cutpoint*. In the experimental group, only 13.33% of the subjects registered a score over the *cutpoint*. The logistic regression model was statistically significant, x^2^(1) = 41.054, *p* < 0.0005 for the experimental group.

For ODI, the MCID score was established as a percentage-based change score from baseline. Different authors have suggested a number of MCIDs represented by a minimal change in scores that have been anchored to a patient’s perception of clinical importance. Minimal clinically important difference change score *cutpoints* for the ODI have been suggested as: 50% change [53]; or 30% change [54]. Success for 30% improvement on the ODI was calculated as [(ODI raw score initial − ODI raw score final)/ODI raw score initial × 100%] ≧ 30%. The dependent variable was dichotomized into successful and unsuccessful groups. Subjects with the baseline score already below the *cutpoint* for success were excluded and the *N* was adjusted accordingly. A logistic regression analysis was completed to account for the differences in sample size between models using the same process, but only including patients with an initial ODI above 30% improvement. For all regression calculations, a *p* value of ≤0.005 was considered significant. 

In the experimental group, 69% from the patients registered a score over the 10-points improvement *cutpoint*. In the experimental group, none of the subjects registered a score over the *cutpoint*. The logistic regression model was statistically significant, x^2^(1) = 9.654, *p* < 0.0005. for the experimental group, and was not statistically significant, x^2^(1) = 1.257, *p* > 0.0005 for the control group

The results of our study suggest that, in the lumbar disc disease, a combined rehabilitation program may be more effective in terms of pain and disability reduction, if it starts early after diagnosis. Similar results were reported by other authors [12,55,56,57] who demonstrated that early intervention displayed significant improvement in pain, disability and socioeconomic outcomes (such as return-to-work and healthcare utilization) [58,59,60].

However, other authors think that the effect of early PT on pain and disability are significant, but the improvements were modest and did not last long [55,56].

A key strength of the present study is the complex rehabilitation treatment with an adapted and individualized physical therapy program, electrotherapy procedures correlated to the symptoms and specific hydrokinetic therapy.

Another strength of this study is that the parameters considered in the study highlight the functional and symptomatologic status of patients.

Clinicians need to take into account that the combination of ET, HT and adherence to an individualized PT program is of great importance for the treatment of LDP.

For patients, it is important to realize that the LDP can be treated conservatively without the need for surgery and that it is very important to consult a doctor as soon as possible in order to prevent further aggravation of the pathology.

It is important for researchers to remember that treatment is individualized according to the patient’s symptoms and clinical condition, and that the PT program must be tailored to the clinical condition of the patients.

This study was limited by the relatively small number of patients and the short duration of the study.

Our findings may stimulate further research with longer follow-up periods and larger patient groups.

## 5. Conclusions

The current study proves the importance of combining ET with HT and PT. Patients who received this treatment combination showed an extremely significant improvement in pain relief, and reduction in functional disability after six months of treatment. The earlier the RT is implemented, the lower the pain perception and level of disability associated with the lumbar disease.

## Figures and Tables

**Figure 1 jpm-12-00741-f001:**
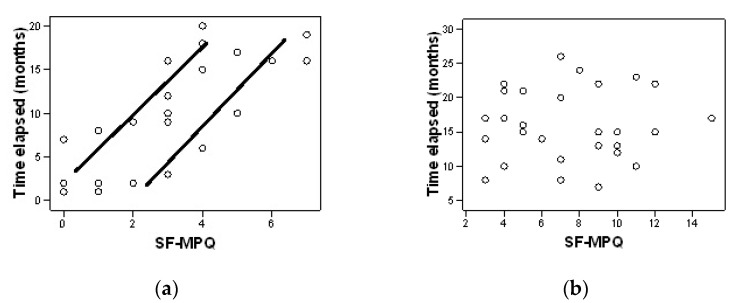
Correlation between the time elapsed from diagnosis and the level of pain after six months for the experimental group (**a**) and control group (**b**).

**Figure 2 jpm-12-00741-f002:**
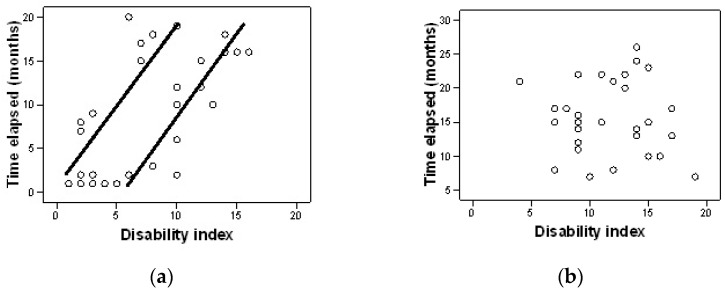
Correlation between the time elapsed from diagnosis and the disability index after six months for the experimental group (**a**) and control group (**b**).

**Table 1 jpm-12-00741-t001:** Subjects’ demographic characteristics (mean ± SD/%), time from diagnosis to first treatment (mean ± SD) and presence of radiculopathy (%) (*n* = 60).

Characteristics	Group A (*n* = 30)	Group B (*n* = 30)	*p*
**Age (years)**	53.30 ± 11.20	52.97 ± 11.04	0.908
**BMI (kg/m^2^)**	29.28 ± 4.71	29.89 ± 4.41	0.685
**Time elapsed (months)**	9.0 ± 6.73	10.63 ± 5.32	0.920
**Gender (%) men** **women**	46.7	36.7	0.432
	53.3	63.3	
**Comorbidities (%) yes** **no**	53.3	60.0	0.602
	46.7	40.0	
**Radiculopathy (%) right** **left** **no**	26.7	33.3	0.717
	30.0	33.3	
	43.3	33.3	

**Table 2 jpm-12-00741-t002:** Comparisons of the initial parameters between the two groups (confidence interval 95%) (mean ± SD).

Parameters	Group A (*n* = 30)	Group B (*n* = 30)	*p* *	95% CI [Lower/Upper]
VAS-M	3.80 ± 1.972	4.07 ± 1.721	0.435	−1.223/0.690
VAS-E	3.17 ± 1.724	3.60 ± 1.567	0.614	−1.285/0.418
SF-MPQ	12.57 ± 6.027	12.77 ± 4.897	0.888	−3.040/2.640
FTF	17.02 ± 10.841	17.57 ± 4.321	0.797	−3.715/4.815
LS	11.20 ± 1.126	12.48 ± 1.334	0.736	−0.692/0.492
ILS	8.38 ± 0.537	8.04 ± 0.666	0.058	0.575/1.185
Right LLF	12.90 ± 3.573	13.81 ± 3.008	0.289	−2.620/0.794
Left LLF	12.62 ± 3.314	13.85 ± 2.948	0.134	−2.852/0.391
LF strength	10.00 ± 5.206	12.63 ± 2.109	0.013	−4.686/−0.581
LE strength	9.80 ± 6.641	13.00 ± 2.101	0.015	−5.745/−0.655
ODQ	17.37 ± 5.605	17.53 ± 4.455	0.899	−2.783/2.450
ODQ (%)	34.73 ± 11.209	35.07 ± 8.909	0.899	−5.566/4.900

VAS-M = Visual Analog Scale (morning); VAS-E = Visual Analog Scale (evening); SF-MPQ = Short Form McGill Pain Questionnaire; FTF = fingertip-to-floor distance; LS = Lumbar Schober test; ILS = Inverted lumbar Schober test; LLF = lateral lumbar flexion; LF strength = strength for flexor muscles; LE strength = strength for extensor muscles; ODQ = Oswestry Disability Questionnaire; * *p* < 0.05.

**Table 3 jpm-12-00741-t003:** Changes expressed as crude values in the pain intensity (VAS and SF-MPQ), lumbar spine mobility (FTF, LS, ILS, Right LLF, Left LLF, LF, LE) and disability level (ODQ and ODQ%) [confidence interval 95%].

	Group A (*n* = 30)		Group B (*n* = 30)		Inter-Action	Effect Size	Group A Changes		Group B Changes	
	Baseline (Mean ± SD)	Post (Mean ± SD)	Baseline (Mean ± SD)	Post (Mean ± SD)	*p*	*p*	*p*	95% CI Lower/Upper	*p*	95% CI Lower/Upper
VAS-M	3.80 ± 1.972	1.30 ± 1.343	4.07 ± 1.721	2.72 ± 1.907	0.002 *	0.163	0.000 *	1.882/3.118	0.001 *	1.111/1.578
VAS-E	3.17 ± 1.724	0.60 ± 0.724	3.60 ± 1.567	2.72 ± 1.701	0.000 *	0.296	0.000 *	1.965/3.169	0.051 *	1.087/1.580
SF-MPQ	12.57 ± 6.027	2.70 ± 2.120	12.77 ± 4.897	7.47 ± 3.137	**0.000 ***	**0.457**	**0.000 ***	7.996/11.671	**0.000 ***	4.521/6.076
FTF	17.02 ± 10.841	11.20 ± 1.126	17.57 ± 4.321	18.12 ± 6.102	**0.005 ***	**0.126**	**0.000 ***	−7.687/−3.940	**0.001 ***	−0.244/1.344
LS	11.20 ± 1.126	13.82 ± 0.932	12.48 ± 1.334	13.92 ± 1.325	**0.000 ***	**0.313**	**0.005 ***	2.277/2.963	**0.005 ***	0.706/1.454
ILS	8.38 ± 0.537	8.93 ± 0.452	8.04 ± 0.666	8.05 ± 0.702	**0.034**	**0.076**	**0.004 ***	0.366/0.734	0.841	0.091/0.111
Right LLF	12.90 ± 3.573	14.13 ± 3.386	13.81 ± 3.008	13.65 ± 3.094	0.571	0.001	**0.001 ***	−1.781/−0.685	0.249	−0.157/−0.517
Left LLF	12.62 ± 3.314	13.48 ± 2.845	13.85 ± 2.948	13.67 ± 3.061	0.821	0.006	**0.005 ***	−1.901/−0.545	0.284	−0.116/0.429
LF strength	10.00 ± 5.206	14.80 ± 5.517	12.63 ± 2.109	12.47 ± 2.501	0.093	**0.071**	**0.001 ***	−5.970/−3.630	0.538	−0.380/0.713
LE strength	9.80 ± 6.641	14.30 ± 7.489	13.00 ± 2.101	13.13 ± 2.763	1.167	0.011	**0.000 ***	−5.676/−3.324	0.670	−0.767/0.500
ODQ	17.37 ± 5.605	8.13 ± 3.785	17.53 ± 4.455	15.60 ± 3.616	**0.001 ***	**0.185**	**0.000 ***	8.026/10.441	**0.000 ***	5.220/6.647
ODI (%)	34.73 ± 11.209	17.93 ± 6.269	35.07 ± 8.909	23.13 ± 7.099	**0.004 ***	**0.135**	**0.000 ***	13.927/19.673	**0.000 ***	10.526/13.341

Group A = experimental group; Group B = control group, VAS-M = Visual Analog Scale (morning); VAS-E = Visual Analog Scale (evening); SF-MPQ = Short Form McGill Pain Questionnaire; FTF = fingertip-to-floor distance; LS = Lumbar Schober test; ILS = Inverted lumbar Schober test; LLF = lateral lumbar flexion; LF strength = strength for flexor muscles; LE strength = strength for extensor muscles; ODQ = Oswestry Disability Questionnaire; * *p*  <  0.05; ***Bold*** states that there is a significant difference.

## Data Availability

The datasets either used, analyzed, or both, during the current study are available from the corresponding authors on reasonable requests.

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
