# Peer review of "Effects of Complex Rehabilitation Program on Reducing Pain and Disability in Patients with Lumbar Disc Protrusion—Is Early Intervention the Best Recommendation?"

_jpm, 2022, doi:10.3390/jpm12050741_

Round 1
Reviewer 1 Report
To authors and editors
This is a very interesting manuscript; thus, it can be published in JPM after some below revisions
1) In the correlation chart, please show crossline.
2) Please show MRI image prior and after therapy.
3) Do you have VAS score for this study. If having, please show result.
4) The figure is not good please make the white fill instead of grey. In addition, the figure is quite small, make it lagger
5) Pre-treatment and after treatment group are needed to compare by Wilcoxon or paired T test instead of Student T test. Please get advice from statisticians.
Sincerely
Reviewer 2 Report
The purpose of this study was to verify if starting an early complex rehabilitation treatment, in Patients affected by LDP, results into a significant better functional outcome at six months of treatment.
Title: it is not clear. Indeed, it seems to suggest that pain and disability increase when the patients were early treated.
Abstract: It is not clear, and it have to rewritten basing on the paper text.
Introduction: The introduction explains the LDP pathological conditions, the CoG modifications and LDP risk factors. On the other hand, this is a paper that aims to investigate if an early complex rehabilitation might results in better outcomes (pain, functionality QoL etc). In my opinion, the introduction has to analyze the state of art in literature about type, intensity and timing for LDP treatment. After, the introduction has to explain what is not clear in literature in relation with the aim of this paper. For this aim, I suggest George SZ, Fritz JM, Silfies SP, Schneider MJ, Beneciuk JM, Lentz TA, Gilliam JR, Hendren S, Norman KS. Interventions for the Management of Acute and Chronic Low Back Pain: Revision 2021. J Orthop Sports Phys Ther. 2021 Nov;51(11):CPG1-CPG60. doi: 10.2519/jospt.2021.0304. PMID: 34719942. This can help you for treatment, and for the literature.
Materials and Methods: Is this a prospective cohort study, or a clinical trial? In my opinion, since there are the experimental group and a control group, this is a randomised clinical trial. On my knowledge, there are not evidence for treating LDP (control group) only with electrotherapy. Do you consider this treatment placebo? On my opinion, if you'd like to investigate if early rehabilitation program might to result in better outcomes, you had to treat patients with the same rehabilitation program starting earlier in the experimental group; or, if you would to investigate if an intensive treatment might result in a better outcome, you have to treat the experimental group, for example, two times each day. Ultimately, the rationale for this type of study design and type of treatment is not clear.
Data analysis: In my opinion, assessments have to be tested with ANOVA for repeated measures, and data analysis by correlation is not needed.
Results and discussion: this have to rewired after the revision of introduction and materials and methods.
Reviewer 3 Report
Thank you for giving me this opportunity to review this article. The article is well written, though I have some serious concerns regarding the article.
Abstract:
- Include the randomization type and allocation procedure in detail.
- Mention the eligibility criteria of the study participants.
- Include the study setting and study duration.
- Mention the intervention procedures.
- Mention the statistical tests used for the study analysis.
- Mention the reports with 95%CI (Upper – lower limit) for all the variables.
- The conclusion should be more concise, self-explanatory, and drawn on the basis of study reports.
Manuscript
- Mention the prevalence rate of lumbar disc prolapse in LBP cases.
- The reference 9 is for general LBP, not for LDP. (not suitable) The treatment approach is totally different for these two clinical cases.
- How come this trial is differing from the references 9, 18-21?
- Give more information about complex physiotherapy program, its application, merits, and demerits.
- Include more recent references regarding the research problem and justify the research gap.
- Include the study hypothesis.
- Include the clinical significance of this study over clinicians, patients, and researchers.
- Include the clinical trial registration number.
- Mention who has diagnosed the condition and is included in the trial?
- Mention the detail eligibility criteria of the study participants.
- Include the type of randomization and allocation procedure in detail.
- Mention the primary and secondary outcome measures and its reliability and validity.
- What is the need of using two pain intensity measurements – MPQ and VAS?
- Give reference for hydrotherapy program?
- What is the need of providing all the electro modalities such as TENS, IFT and Magneto therapy for the patients? – Justify.
- Mention the blinding procedure in detail.
- Include the detail description of statistical tests used for analysis (association between variables and time).
- Mention the detailed demographic description of the study participants in the results session (Duration of injury, associated problems, etc...)
- In the results section, please discuss the treatment compliance rate, adverse effects, and the number of dropouts with reason.
- Mention the reports with 95%CI (Upper – lower limit) for all the variables.
- Include the effect size of each variable between the groups.
- Find the MCID score and discuss it in the discussion session.
- Mention the mechanism (physiological and biochemical) behind the intervention over LBP patients with recent references.
Author Response
Please see the attachment Response Reviewer 3

Round 2
Reviewer 3 Report
Thank you for giving me this opportunity to review this article. The article is well written, though I have some serious concerns regarding the article.
Abstract:
- Present the abstract with specific sub-titles like background, objectives, methods, results, etc.
- The results should be presented with 95%CI (upper limit – lower limit) for all the variables.
- The conclusion should be concise and self- explanatory and drawn on the basis of study reports.
Manuscript
- Include the clinical significance of this study over clinicians, patients, and researchers after the study hypothesis.
- As per ICMJE guidelines, it is mandatory to register with a clinical trial registry, if the study involves human subjects with prospective recruitment.
- Please follow the CONSORT guidelines to present the study. – please attach the CONSORT checklist.
- Mention the randomization and allocation procedure in detail.
- Mention who has diagnosed the condition and is included in the trial?
- How the sample size is calculated and its primary reference?
- Mention the blinding information.
- Include the details of treatment adherence, adverse effects, and dropout information.
- Include the effect size of all the variables.
- Mention the MCID value of the primary variable and discuss the mechanism behind it.
- Include the strength of the study.
Author Response
I attach the response to reviewer 3 round 2

Round 3
Reviewer 3 Report
- Make the introduction part into three to four paragraphs.
- Present the methods section as per the CONSORT guidelines title and sub-titles.
- Please justify the need for TENS, IFT, and magnetic field therapy for the patients.
Author Response
Response to Reviewer 3 – Review Report (Round 3) Comments
All the responses have been made by track changes in the main document.
Manuscript
1. Make the introduction part into three to four paragraphs.
We made the modifications.
2. Present the methods section as per the CONSORT guidelines title and sub-titles.
We made the modifications.
3. Please justify the need for TENS, IFT, and magnetic field therapy for the patients.
We have included the justification in the manuscript. (rows 164 - 166)
In order to address the following recommendation: „English language and style are fine/minor spell check required” – we reviewed the manuscript and made the corrections
